# From Graph Embedding to LKH: Bridging Learning and Heuristics for a Streamlined General TSP Solver

## Abstract

The Traveling Salesman Problem (TSP) is known as one of the most notorious NP-hard combinatorial optimization problems. In recent decades, researchers from fields such as computer science, operations research, and artificial intelligence including deep learning (DL) have made numerous attempts on the problem. Among the works, the Lin-Kernighan-Helsgaun (LKH) heuristic algorithm is one of the most competent methods for obtaining optimal or near-optimal solutions. Despite the rapid development in DL-based solvers, few of them can defeat LKH in terms of both running efficiency and solution quality across different distributions. In this paper, we would introduce a very novel approach that enhances LKH with graph embedding (GE) techniques in solving general TSP (distances can be non-metric and asymmetric), named as Embed-LKH. It is presented as two stages: i) in the GE stage, it transforms the distances to transition probabilities, then conduct GE given the transition probabilities, and finally it uses the learned embeddings to construct the so-called 'ghost distances'; ii) in the LKH stage, LKH generates candidates based on the ghost distances but searches tours according to the original distances.

As the experiments show, compared with the original LKH counterpart, in most cases, our approach can obtain better solutions within the same amount of trials across six distance distributions (non-metric and asymmetric: normal, uniform, exponential, metric and symmetric: Euclidean 2D/10D/50D) and two problem scales (TSP-100/1000). The source files, running scripts, and data are in the anonymous link https://anonymous.4open.science/r/EmbedLKH-BF80/, which will be made publicly available after the review.

*If I have seen further, it is by standing on the shoulders of giants.*

– Isaac Newton.

## 1 Introduction

The Travelling Salesman Problem (TSP) is one of the most well-known NP-hard combinatorial optimization (CO) problems. Given a set of nodes and the distances between them, TSP aims to find the shortest tour that visits each node exactly once and returns to the starting node. With applications spanning logistics, telecommunications, and genetics, TSP plays a crucial role in route optimization and network planning (Applegate et al., 2007; Rardin & Rardin, 1998; Matai et al., 2010).

**General TSP and its Practical Significance.** In this paper, we study the *general TSP*, where the costs from node to node can be non-Euclidean (are not distances derived from Euclidean coordinates) and asymmetric (the cost from node $i$ to $j$ does not have to be equal to that from $j$ to $i$). Compared with the symmetric TSP defined by 2D coordinates which is much more commonly studied by literature (Kool et al., 2018; Kwon et al., 2020; Li et al., 2023; Qiu et al., 2022), the general TSP is more ubiquitous in practical applications: For example, the time it takes for vehicles to travel between two places does not entirely depend on the physical distance between the two places, but is highly affected by road congestion and speed limits (65 mph on highway but 25 mph in residential districts). That leads to a non-Euclidean time cost. Meanwhile, the time cost can be also asymmetric due to different congestion situations in different directions (for example, during morning rush hours, the required time of traveling from residential areas to work areas is much higher than the opposite direction), as well as probably different road conditions (some roads are one-way).

**Research Progress.** One algorithm that cannot go unmentioned in the context of the TSP is LKH (Lin-Kernighan-Helsgaun (Helsgaun, 2000; 2017)), widely regarded as one of the most effective heuristic algorithms for TSP due to its near-optimal solutions, high speed, and good scalability. Huge success has been made in neural solvers for TSP in 2D Euclidean space. Notably, some of the works (Sun & Yang, 2023; Li et al., 2024) achieve a comparable performance with LKH. However, on general TSP, there remains a considerable gap between cur-

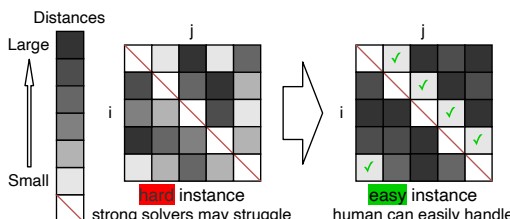

Figure 1: Motivation: reducing the difficulty of the input instances helps solvers more easily find better solutions.

rent neural solvers and LKH. In this paper, we are going to bridge this gap by innovatively utilizing graph embedding to enhance LKH, which falls in the so-called **Embed-LKH**. And for the first time, Embed-LKH outperforms LKH on the general TSP (including the Euclidean ones) within the same amount of trials. Just as we quoted Newton in the preface, we believe that the shortcut to surpassing LKH is to stand on LKH's shoulder, and as shown by experiments, our Embed-LKH does see further.

**Motivation.** The motivation is straightforward: to reduce the difficulty of input TSP instances. As illustrated in Fig. 1, there are hard cases and easy cases. For hard instances, even strong solvers like LKH struggle to find the optimal solution, whereas for easier instances, a simple greedy algorithm (e.g., nearest neighbor) suffices. Our objective is to transform a difficult instance into an easier one that likely shares the same optimal solution, allowing LKH to find a shorter tour with fewer trials. Importantly, while we adjust the distances during the transformation, the tour length is always computed using the original distances of the hard instance.

We notice that the graph embedding (GE) techniques (e.g., Perozzi et al. (2014); Grover & Leskovec (2016)), originally developed for graph mining tasks (e.g., community discovery, social link prediction, etc.), possess remarkable properties that could be highly effective for general TSP solving, including: **i) Scalability.** With a single machine of 128GB memory, GE methods (e.g., Tang et al. (2015); Qiu et al. (2019a)) can easily deal with graphs of millions of nodes. **ii) Unsupervised learning.** Most GE methods work in an unsupervised style, making it particularly suitable for the general TSP, where optimal solutions are usually inaccessible as supervisory information. **iii) Applicability on non-attributed graphs**. GE directly learns from graph topology without node attributes. These virtues of GE also motivate us to apply GE in the challenge of solving general TSP.

**Our Techniques and Highlighted Features.** In our Embed-LKH, we first apply GE techniques on the transition probability matrix transformed from the distance matrix of the general TSP. This allows us to construct a 'ghost distance matrix', a modified distance matrix that we expect will be easier to solve compared to the original distances. We then generate candidates based on the ghost distances while searching tours according to the original distances to ensure the accuracy of the computed tour length. Below, we summarize the main virtues of Embed-LKH:

- **Effectiveness.** We investigate six distance distributions including three non-metric and asymmetric (normal, uniform, exponential), and three metric and symmetric (Euclidean 2D/10D/50D). Results show that in most cases Embed-LKH can outperform learning-based solvers, as well as the original LKH within the same number of searching trials.
- **High efficiency and scalability.** We utilize closed-form solution for GE, avoiding gradient descent thus achieving higher efficiency of the GE steps. We further propose an asynchonization scheme for Embed-LKH where GE steps and the LKH step are conducted in different threads asynchronously. These technical points endow Embed-LKH with high efficiency and good scalability.
- **One-shot.** Unlike popular learning-based (Kool et al., 2018; Qiu et al., 2022; Li et al., 2023) and learning-enhanced heuristic (Hudson et al., 2021; Xin et al., 2021) methods for solving TSP, Embed-LKH runs in the one-shot manner, allowing it to be applied to instances of different scales without a typical resource-consuming pre-training stage, while also avoiding generalization issues from training data to testing data.
- **Invariance to perturbations.** Most supervised (Joshi et al., 2019; Fu et al., 2021) or reinforcement learning-based methods (Kool et al., 2018; Qiu et al., 2022; Kwon et al., 2021) suffer from performance collapse due to the fragility of neural predictions which can be immensely affected by any perturbations on the input distances, largely hindering their applicability to real-world

cases. As we prove in Sec. 6, Embed-LKH is invariant to row-wise distance disturbances and node permutations which would not change the optimality of the solutions.

## 2 RELATED WORKS

For space limit, we put related works in Appendix B where we detailedly analyze the differences between our work and others and summarize the main points in Table 5.

## 3 BACKGROUNDS AND PRELIMINARIES

We give a formal definition of our studied general TSP in Sec. 3.1. Then in Sec. 3.3, we give preliminaries of the Graph Embedding (GE) techniques. We will elaborate on how to bridge the gap of the two seemingly totally irrelevant worlds in the last.

### 3.1 GENERAL TSP DEFINED BY DISTANCE MATRICES

**Definition 1 (General Traveling Salesman Problem).** *Given a node set $\mathcal{V}$ ($\mathcal{V} = \{1, 2, \cdots, N\}$) along with a distance matrix $\mathbf{D}_{i,j} \in [0, +\infty)^{N \times N}$ where the entry $\mathbf{D}_{i,j}$ is the distance from node $i$ to $j$, the problem is to find the tour (a Hamiltonian cycle that visits all the nodes exactly once and returns to the starting node) $\tau = (\tau_1, \cdots, \tau_N, \tau_1)$ to minimizes the cost $\sum_{i=1}^{N-1} \mathbf{D}_{\tau_i, \tau_{i+1}} + \mathbf{D}_{\tau_N, \tau_1}$. Without losing generality or decreasing the hard level of the problem, we assume that $\mathbf{D}_{ij}$ is a positive value. In this paper, we consider TSP in the general cases, namely*

- ***Asymmetric.** The symmetry, i.e. $\mathbf{D}_{i,j} = \mathbf{D}_{j,i}$, does NOT have to hold.*
- ***Non-Metric.** The triangle inequality, i.e. $\mathbf{D}_{i,j} + \mathbf{D}_{j,k} \geq \mathbf{D}_{i,k}$, does NOT have to hold.*

### 3.2 LKH ALGORITHM

LKH is a highly encapsulated tool tailored for vehicle routing problems including TSP with abundant of technical tricks in it. Diving deep into the working mechanisms of LKH is not the purpose of the paper. Readers who are interested in the technical details of LKH may refer to Appendix C. We highly recommend readers to regard LKH as a black box which works by two steps: 1) LKH_GeneCand($\mathbf{D}$): Given a distance matrix $\mathbf{D}$, it generates *candidates* for each node. Here 'candidates' are indicating how likely the edge from the node to a candidate shows in the final solution; 2) LKH_SearchTour($\mathbf{D}, Candidates$): Given distance matrix $\mathbf{D}$, it searches tours based on the generated candidates.

### 3.3 WORD2VEC-BASED GRAPH EMBEDDING TECHNIQUES

**Technical Overview of Word2vec.** Word2vec (Mikolov et al., 2013) is one of the most well-known algorithms in NLP, winning NeurIPS 2023 Test of Time Awards. It is a shallow neural network model that learns to represent words in a continuous vector space from a given corpus. The training objective of Skip-gram model in word2vec is to predict surrounding *context words* for the *center words*. Word2vec assigns each word $i$ one word embedding $\mathbf{x}_i \in \mathbb{R}^d$, and one hidden embedding $\mathbf{h}_i \in \mathbb{R}^d$. From the natural language text corpus, we can obtain the *empirical co-occurence probability* that a word $j$ shows as the context word of another center word $i$, denoted by $p(j|i)$, and the frequency that the word $i$ is selected as the center word, denoted by $p(i)$. Then we have the objective:

$$\text{maximize} \quad J := \sum_{i,j \in \mathcal{V}} p(i)p(j|i) \log \frac{\exp \mathbf{x}_i^\top \mathbf{h}_j}{\sum_{j' \in \mathcal{V}} \exp \mathbf{x}_i^\top \mathbf{h}_{j'}}, \tag{1}$$

where $\mathcal{V}$ is the vocabulary (minor notation abuse with node set $\mathcal{V}$ of similar meaning). Notice that the denominator of the objective Eq. 1 is too computationally expensive to be practical. To this end, the negative sampling technique is proposed (Mikolov et al., 2013), whose objective is defined as:

$$\text{maximize} \quad J := \sum_{i,j \in \mathcal{V}} p(i)p(j|i) \Big[ \log \sigma(\mathbf{x}_i^\top \mathbf{h}_j) + \sum_{s=1}^{S} \mathbb{E}_{j' \sim \mathbb{P}_{neg}} \log \sigma(-\mathbf{x}_i^\top \mathbf{h}_{j'}) \Big], \tag{2}$$

where $S$ is the number of negative samples for each positive sample term $\log \sigma(\mathbf{x}_i^\top \mathbf{h}_j)$ in the objetive, and $\mathbb{P}_{neg}$ is an empirical distribution of negative samples.

**Word2vec-Based GE.** Similar to word embedding that aims to represent words as vectors, given an input graph $G = (\mathcal{V}, \mathcal{E})$ ($\mathcal{V}$: nodes, $\mathcal{E}$: edges), GE aims to represent the nodes of as vectors for further node-level downstream tasks. Word2vec-based GE methods (e.g., Perozzi et al. (2014); Grover & Leskovec (2016)) first design a random walk strategy (where $p(i)$ and $p(j|i)$ are explicitly

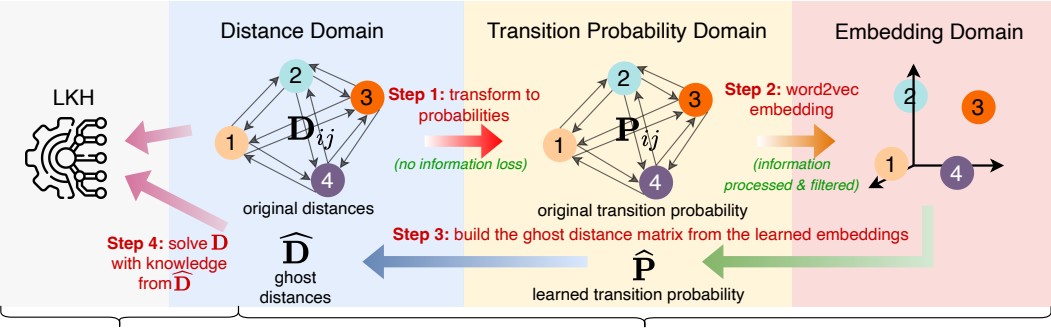

Figure 2: The 4-step workflow of Embed-LKH which contains three GE steps and one LKH step.

or implicitly defined) to convert the graph-structure data into node sequence data, and then feed the node sequences to word2vec just like dealing with the natural language corpus, after which we would obtain the node embeddings. From this perspective, these word2vec-based GE methods are essentially the art of designing $p(i)$ and $p(i|j)$ for the input graph $G$. We write such a procedure as the following formula:

$$\mathbf{X} := [\mathbf{x}_i]_{i=1}^N, \ \mathbf{H} := [\mathbf{h}_i]_{i=1}^N \leftarrow \texttt{word2vec}\Big(\mathcal{V}, p(i), p(j|i)\Big) \tag{3}$$

where $\mathbf{X}, \mathbf{H} \in \mathbb{R}^{d \times N}$ are the node embedding matrix and hidden embedding matrix respectively.

So far, we've already introduced basic techniques of word2vec-based GE. The key step of applying the techniques to general TSP, is to define $p(i)$ and $p(j|i)$ for a given distance matrix $\mathbf{D}$. We will explain the methodology in detail in the next section.

## 4 EMBED-LKH: GRAPH-EMBEDDING-ENHANCED LKH

### 4.1 METHOD OVERVIEW AND NOTATION EXPLANATIONS

As presented in the backgrounds of last section, GE learns from $p(i)$ and $p(j|i)$. So, **step 1** is to transform $\mathbf{D} \in [0, +\infty)^{N \times N}$ in the distance space to a *transition probability* matrix $\mathbf{P} \in [0, 1]^{N \times N}$ in the probability space, whose element $\mathbf{P}_{ij}$ is the transition probability from node $i$ to $j$. **Step 2** is conducting a word2vec-based GE for $\mathbf{P}$. **Step 3** is to construct a *ghost distance* matrix $\widehat{\mathbf{D}}$ which is supposed to be an easier case than original distance matrix $\mathbf{D}$. And the final **step 4** is running LKH to solve $\mathbf{D}$ but using candidates generated from $\widehat{\mathbf{D}}$.

The method framework is illustrated by Fig. 2, with the complete algorithm presented in Alg. 1 where we give detailed analysis of time complexity of each step. As the supplementary for easy reading, we give the notation list in Table 4, Appendix A.

### 4.2 STEP 1: TRANSFORM DISTANCES TO TRANSITION PRBABILITY

**The Relation between Distances and Probabilities.** We consider such a *path*[1] $\pi = (\pi_1, \pi_2, \ldots, \pi_M)$ of $M$ nodes. In the distance space, the length of $\pi$ is defined by the sum of the distances between adjacent nodes in $\pi$. While in the probability space, the probability $p(\pi)$ of formulating such a $\pi$ can be given by cumulative product of the transition probabilities $\mathbf{P}_{\pi_i \pi_{i+1}}$ between adjacent nodes $\pi_i$ and $\pi_{i+1}$ in the path. The differences can be mathematically shown by the below Eq. 4:

$$\text{length (for TSP): } L(\pi) = \sum_{i=1}^{M-1} \mathbf{D}_{\pi_i, \pi_{i+1}}, \quad \text{probability (for GE): } p(\pi) = \prod_{i=1}^{M-1} \mathbf{P}_{\pi_i, \pi_{i+1}}. \tag{4}$$

**Methodology.** Based on the observation above, we can define a simple way to transform $\mathbf{D}$ to $\mathbf{P}$ with a row-wise softmax function over $-\mathbf{D}$ as below:

$$(\text{step 1}) \quad \mathbf{P}_{i,j} \leftarrow \texttt{Softmax}(-\mathbf{D}_{i,:})_j := \frac{\exp(-\mathbf{D}_{i,j})}{\sum_{j'=1}^N \exp(-\mathbf{D}_{i,j'})}, \tag{5}$$

By Eq. 5, we can see that the smaller $\mathbf{D}_{i,j}$ is, the higher transition probability $\mathbf{P}_{i,j}$ will be, which is in accordance with the intuition of TSP. More strictly, Eq. 5 also ensures the following Proposition 1:

---

[1]Notation clarification: a tour $\tau$ is a special type of paths $\pi$ where the starting node is also the last node and other nodes show exactly once, see Definition 1.

---

**Algorithm 1** Embed-LKH: Graph-Embedding-Enhanced LKH.

---

1: **Input:** Distance matrix $\mathbf{D} \in \mathbb{R}^{N \times N}$;
2: **Parameters for GE**: embedding dimension $d$, number of negative samples for each node $K$.
3: **Parameters for LKH**: the number of runs #runs, the number of maximum trial times #max_trials.
4: ► **Step 1:** Distance transformation (Sec. 4.2). **Time:** $O(N^2)$.
5: $\mathbf{P} \leftarrow diag(\exp(-\mathbf{D})\mathbf{1})^{-1} \exp(-\mathbf{D})$  ► Transform from distance space to probability space
6: ► **Step 2:** Graph embedding (Sec. 4.3 and 5.1).
7: $\mathbf{Q} \leftarrow f_p(\mathbf{P})$  **Time:** $O(N^3 W)$ for random walk, $O(N^2 \log N)$ for sparsification (optional);
8: $\mathbf{U}_d, \mathbf{\Sigma}_d, \mathbf{V}_d^\top \leftarrow \text{SVD}\big(\log(N\mathbf{Q}/K); d\big)$; $\mathbf{X} \leftarrow \sqrt{\mathbf{\Sigma}_d}\mathbf{U}_d^\top$; $\mathbf{H} \leftarrow \sqrt{\mathbf{\Sigma}_d}\mathbf{V}_d^\top$  **Time:** $O(N^2 \log d + Nd^2)$
9: ► **Step 3:** Build ghost distance matrix(Sec. 4.4).
10: $\widehat{\mathbf{P}} \leftarrow \frac{K}{N} \exp(\mathbf{X}^\top \mathbf{H})$; $\widehat{\mathbf{P}} \leftarrow \widehat{\mathbf{P}}/\text{sum}(\widehat{\mathbf{P}}, \dim = 1)$ ► Reconstruct the transition probability by embeddings
11: $\widehat{\mathbf{D}} \leftarrow -\log(\widehat{\mathbf{P}})$; $\widehat{\mathbf{D}} \leftarrow \widehat{\mathbf{D}} - \min(\widehat{\mathbf{D}})$  ► Build ghost distance matrix $\widehat{\mathbf{D}}$
12: ► **Step 4:** Run LKH with candidates generated from ghost distances (Sec. 4.5)
13: **Time:** $O(N^3)$ for line 14 step and $O(N \text{ #runs} \cdot \text{#max\_trials} \cdot \text{#cand})$ for line 15, the same as original LKH
14: $Cand \leftarrow \text{LKH\_GenCand}(\widehat{\mathbf{D}})$  ► Generate candidates from ghost distance matrix
15: $\tau^* \leftarrow \text{LKH\_SearchTour}(\mathbf{D}, Cand)$  ► Search for a best tour $\tau^*$ from candidates,

---

**Proposition 1** (Distance $\mapsto$ Transitional Probability). *We consider a TSP instance attached with a distance matrix* $\mathbf{D}$. *We then define the transition probability* $\mathbf{P}$ *by Eq. 5. Observing that the length and probability of a path* $\pi$ *defined in Eq. 4 also fit a tour* $\tau$, *we have:*

*1. For two tours $\tau$ and $\tau'$, if $L(\tau) < L(\tau')$, then $p(\tau) > p(\tau')$.*
*2. The tour $\tau$ of the shortest length $L(\tau)$ is also the tour that has the highest probability $p(\tau)$.*

*Proof.* Substituting $\pi$ in Eq. 4 with $\tau$, we have

$$p(\tau) = \frac{\exp(-\mathbf{D}_{\tau_N \tau_1})}{\sum_{j'=1}^N \exp(-\mathbf{D}_{\tau_N j'})} \prod_{i=1}^{N-1} \frac{\exp(-\mathbf{D}_{\tau_i \tau_{i+1}})}{\sum_{j'=1}^N \exp(-\mathbf{D}_{\tau_i j'})} = \frac{\exp\big(-L(\tau)\big)}{\prod_{i=1}^N \left[ \sum_{j'=1}^N \exp(-\mathbf{D}_{\tau_i j'}) \right]}, \quad (6)$$

whose denominator is a constant that is independent of $\tau$ and only related to $\mathbf{D}$, so $p(\tau)$ monotonically decreases with respect to $L(\tau)$. Then the first proposition holds. And the second proposition is obvious with the first proposition proven. $\square$

Proposition 1 ensures that the transformation from distance space to probability space by Eq. 5 does not compromise the optimality of the tour, while enabling GE at the same time.

### 4.3 STEP 2: GRAPH EMBEDDING IN THE TRANSITIONAL PROBABILITY DOMAIN

Recall in Sec. 3.3, we said that word2vec-based GE is the art of designing $p(i)$ and $p(j|i)$. Therefore, the main content of this step is to design a strategy $f_p : \mathbf{P} \mapsto p(i), p(j|i)$, so that we can obtain embeddings $\mathbf{X}$ and $\mathbf{H}$ by the below formula following Eq. 1:

$$(\text{step 2}) \quad \mathbf{X}, \mathbf{H} \leftarrow \text{word2vec}\Big(\mathcal{V}, f_p(\mathbf{P})\Big). \quad (7)$$

**A trivial strategy:** $p(i) = 1/N$, $p(j|i) = \mathbf{P}_{i,j}$ for large instances which saves running time.

**Random-walk-based Strategy.** Borrowing ideas from deepwalk (Perozzi et al., 2014), we set $p(i) = 1/N$ and $p(j|i) = (\sum_{w=1}^W \frac{1}{w} \mathbf{P}^w)_{ij}$, where $W$ is the 'window size', representing how many hops of information will be integrated. The coefficient $\frac{1}{w}$ indicates that the weight of $w$-hop information reduces as the hop increases. Excessive focus on the one-hop distance between nodes may be detrimental to solving the TSP. Random walk helps to 'see further'.

**Sparsification for Large Instances.** Before random walk, we select top-$K$ probabilities for each node and mask the rest. It helps to filter out many edges that are unlikely to be candidates in the solution. Note that thought sparsification may improve results significantly, it can also be time-consuming.

### 4.4 STEP 3: CONSTRUCT GHOST DISTANCES FROM EMBEDDINGS

After obtaining the embeddings $\mathbf{X}$ and $\mathbf{H}$, we construct the learnt transition probability matrix $\widehat{\mathbf{P}}$ by

$$(\text{step 3.1}) \quad \widehat{\mathbf{P}}_{i,j} \longleftarrow \frac{\exp(\mathbf{x}_i^\top \mathbf{h}_j)}{\sum_{j'=1}^N \exp(\mathbf{x}_i \mathbf{h}_{j'})}, \quad (8)$$

and then construct the '*ghost distance*' matrix $\widehat{\mathbf{D}}$ by a similarly reverse procedure of Eq. 5:

$$\text{(step 3.2)} \quad \widehat{\mathbf{D}} \leftarrow -\log(\widehat{\mathbf{P}}), \text{ then } \widehat{\mathbf{D}} \leftarrow \widehat{\mathbf{D}} - \min(\widehat{\mathbf{D}}), \tag{9}$$

where the second formula aims to set the minimum value of distances as 0. Just as Proposition 1, the procedure of building such a ghost distance matrix from the learnt transitional probability does not hurt the optimality of the tour, as formally presented in Proposition 2. For page limit, the proof is put in Appendix D. Proposition 1 and 2 indicate that the step 1 and 3 will not cause information loss.

**Proposition 2** (Transitional Probability $\mapsto$ Distance, Reverse to Proposition 1)**.** *We consider the transition probability matrix $\widehat{\mathbf{P}}$ between a set of nodes $\mathcal{V}$, with the probability of a path $\pi$ defined by $\hat{p}(\pi) = \prod_{i=1}^{M-1} \widehat{\mathbf{P}}_{\pi_i, \pi_{i+1}}$ (as Eq. 4). We construct a distance matrix $\widehat{\mathbf{D}}$ by as Eq. 9, then we have:*

*1. For two tours $\tau$ and $\tau'$, if $p(\tau) < p(\tau')$, then $\hat{L}(\tau) > \hat{L}(\tau')$ holds in TSP.*
*2. The tour $\tau$ that has the highest probability $\hat{p}(\tau)$ also has the shortest length $\hat{L}(\tau)$ in TSP.*

### 4.5 STEP 4: RUN LKH WITH CANDIDATES GENERATED FROM GHOST DISTANCES

In step 3 we have obtained ghost distances $\widehat{\mathbf{D}}$, which we hope is an easier case to solve than original distances $\mathbf{D}$. And in Sec. 3.2 we've introduced how LKH works by two functions `LKH_GenCand` and `LKH_SearchTour`. In Embed-LKH, we use the learned ghost distances $\widehat{\mathbf{D}}$ to generate candidates, then search tours with $\mathbf{D}$, just as blow:

$$\text{(step 4)} \quad Cand \leftarrow \texttt{LKH\_GenCand}(\widehat{\mathbf{D}}), \quad \tau \leftarrow \texttt{LKH\_SearchTour}(\mathbf{D}, Cand). \tag{10}$$

Since we compute tour length according to $\mathbf{D}$, need to accurately calculate the length of the tour to ensure that the output tour is the shortest in the original distance space but not the one in the ghost distance space.

## 5 TECHNICAL DETAILS FOR HIGHER EFFICIENCY

### 5.1 CLOSED-FORM SOLUTION OF WORD2VEC BASED ON MATRIX FACTORIZATION

In Sec. 3.3, we've introduced the objective of word2vec (Eq. 1). As proven by previous works (Levy & Goldberg, 2014; Qiu et al., 2018), once $p(i)$ and $p(j|i)$ are explicitly known and no other constraints are introduced during the optimization, the embeddings that are learned from the objective can be optimally obtained via matrix factorization. Compared with the gradient descent-based optimizers, e.g., SGD, the matrix factorization-based optimization is much more efficient.

**Methodology.** We first define $y_{ij} := \mathbf{x}_i^\top \mathbf{h}_j$, then we calculate the partial derivative of $J$ with regard to $y_{ij}$ as follows (note that $y_i j$ show not only as the positive sample term, but also as the negative sample term for other positive sample terms):

$$\frac{\partial J}{\partial y_{ij}} = -p(i)\big[p(j|i)\sigma(-y_{ij}) + S\mathbb{P}_{neg}(j)\sigma(y_{ij})\big]. \tag{11}$$

By setting $\frac{\partial J}{\partial y_{ij}}$ to 0 and using the uniform distribution i.e. $\mathbb{P}_{neg}(j') = 1/N$ for negative sampling (which means all the nodes treated equally as negative samples), we would obtain the optimal solution $y_{ij}^* = \log \frac{p(j|i)}{S\mathbb{P}_{neg}(j)} = \log \frac{Np(j|i)}{S}$. We write $p(j|i)$ in the matrix form as $\mathbf{Q}$ with $\mathbf{Q}_{i,j} = p(j|i)$. Then word2vec is indeed conducting the following matrix factorization from left to right:

$$\log(N\mathbf{Q}/S) \approx \mathbf{X}^\top \mathbf{H}. \tag{12}$$

To obtain the optimal $\mathbf{X}$ and $\mathbf{H}$, we can conduct Singular Value Decomposition (SVD) over the LHS term of Eq. 12, and then preserve the highest $d$ singular values and corresponding vectors and construct embeddings $\mathbf{X}$ and $\mathbf{H}$ just as follows:

$$\mathbf{U}_d, \mathbf{\Sigma}_d, \mathbf{V}_d^\top \leftarrow \texttt{SVD}(\log(N\mathbf{Q}/S); d), \text{ then } \mathbf{X} \leftarrow \sqrt{\mathbf{\Sigma}_d}\mathbf{U}_d^\top \text{ and } \mathbf{H} \leftarrow \sqrt{\mathbf{\Sigma}_d}\mathbf{V}_d^\top. \tag{13}$$

**Randomized SVD.** An ordinary procedure of SVD may consume a $O(N^3)$ time. In Embed-LKLH, we have $d \ll N$ (in experiments we set $d < 5$). In this case, randomized SVD (Halko et al., 2009) with a $O(N^2 \log d + Nd^2)$ time complexity can save much time. Randomized SVD for a given matrix $\mathbf{A}$ is conducted by two stage: i) Compute an approximate basis for the range of $\mathbf{A}$. The goal is to obtain a matrix $\mathbf{B}$ to make $\mathbf{A} \approx \mathbf{BB}^*$. ii) Use $\mathbf{B}$ which is much smaller than $\mathbf{A}$ to compute matrix factorization. Readers who are interested in details are recommended to refer to the official paper.

## 5.2 ASYNCHRONIZE EMBED-LKH

Recall that Embed-LKH has 4 steps, including 3 GE steps (step 1-3) and an LKH step (step 4). To improve running efficiency of Embed-LKH and fully maximize CPU utilization, we asynchronize the GE steps and the LKH step by two separate threads as illustrated in Fig. 3. In this way, when dealing with a batch of tasks, the additional

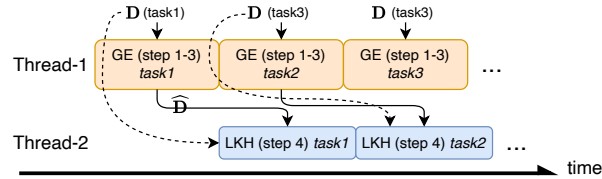

Figure 3: Asynchronization of Embed-LKH.

computation overheads introduced by the GE steps can be significantly mitigated in terms of their impact on the program's runtime efficiency. Go a further step, we can simply increase the number of threads as plotted in Fig. 3 to achieve a higher solving speed. In experiments we use 8 threads for the LKH step and 8 threads for the GE steps as the default.

## 6 THEORETICAL DISCUSSION

### 6.1 INVARIANCE TO THE INPUT PERTURBATIONS

Abundant works for neural TSP solvers deal with the distance perturbations that would not change the optimal solutions by data augmentation (Jiang et al., 2022; Kwon et al., 2020; 2021; Ma et al., 2021). In comparison, Embed-LKH is naturally invariant to some certain perturbations, as presented by the following Proposition 3 and 4.

**Proposition 3 (Row-wise Disturbance Invariance).** *We define a random row-wise disturbance vector as $\mathbf{b} \in \mathbb{R}^N$, whose entry $\mathbf{b}_i$ represents the disturbance on node $i$'s distances to other nodes. If we disturb the input distance matrix $\mathbf{D}$ by $\mathbf{D}' \leftarrow \mathbf{D} + \mathbf{b}$ ($\mathbf{D}'_{i,j} \leftarrow \mathbf{D}_{i,j} + \mathbf{b}_i$), one can easily prove that such a disturbance would not change the optimality of solutions. We conclude that the GE outputs, embeddings $\mathbf{X}$ and $\mathbf{H}$, are also invariant to such a disturbance.*

*Proof.* We find that the disturbance works in step 1. For the new distances $\mathbf{D}'$, we denote the transition probability as $\mathbf{P}'$. By step 1 (Eq. 5), we have:

$$\mathbf{P}'_{i,j} = \texttt{Softmax}(-\mathbf{D}'_{i,:}) = \frac{\exp(-\mathbf{D}_{i,j} - \mathbf{b}_i)}{\sum_{j'=1}^{N} \exp(-\mathbf{D}_{i,j'} - \mathbf{b}_i)} = \frac{\exp(-\mathbf{D}_{i,j})}{\sum_{j'=1}^{N} \exp(-\mathbf{D}_{i,j'})} = \mathbf{P}_{i,j}. \quad (14)$$

So, the disturbance would not change the output of step 1, and also has no influence on the final embeddings $\mathbf{X}$ and $\mathbf{H}$. □

**Proposition 4 (Node Permutation Invariance).** *We define a random permute matrix $\mathbf{M} \in \{0,1\}^{N \times N}$ where each row and each column has exactly one non-zero element. A random permutation over the distance matrix by $\mathbf{D}' \leftarrow \mathbf{M}\mathbf{D}\mathbf{M}^\top$ would not change the optimal solution. We conclude that node permutation does not affect the output solution generated by Embed-LKH, either.*

*Proof.* It is obvious that, by step 1 (Eq. 5), we have the transition probability after permutation $\mathbf{P}' = \mathbf{M}\mathbf{P}\mathbf{M}^\top$; then by step 2 (Eq. 12 and 13), we have $\mathbf{X}' = \mathbf{M}\mathbf{X}$ and $\mathbf{H}' = \mathbf{M}\mathbf{H}$; then by step 3 (Eq. 8 and 9), we have $\widehat{\mathbf{D}}' = \mathbf{M}\widehat{\mathbf{D}}\mathbf{M}^\top$. Here we find that the ghost distance matrix $\widehat{\mathbf{D}}'$ is exactly a randomly permuted $\widehat{\mathbf{D}}$. So for the same nodes of different IDs in $\widehat{\mathbf{D}}$ and $\widehat{\mathbf{D}}'$, the candidates generated by `LKH_GenCand` would be the same. Also, the function `LKH_SearchTour` is irrelevant with node orders. So, the solution given by Embed-LKH would not be affected by node permutation. □

### 6.2 EMBED-LKH IS A ONE-SHOT SOLVER

As we presented in Alg. 1, before testing, Embed-LKH is not trained on a training dataset which contains many instances, but conducted right on the testing instance. So, Embed-LKH is a one-shot solver. The 'one-shot' property endows the method more other virtues including: **i) All-scale applicability.** Unlike previous methods that require ensuring that the size of the test data is the same as the size of the instances in the training dataset, our method can be applied to problems of any scale (as long as the machine has enough memory). **ii) Low training cost.** Unlike previous deep learning methods (e.g., Kwon et al. (2021)), Embed-LKH does not require a significant amount of time for pre-training and has no requirements for hardware (e.g., GPUs). **iii) Generalizability.** Embed-LKH does not have the issue of generalization from training data to testing data.

# 7 EXPERIMENTS

## 7.1 EXPERIMENTAL SETUP

**Hardware.** Experiments for neural methods are conducted on a single NVIDIA RTX3090 24GB GPU with AMD 3970X 32-Core CPU. Our Embed-LKH and other heuristics (LKH, EAX, etc.) are conducted on a machine with Intel Xeon W-3175X CPU and 128GB memory.

**Baselines. Neural solver: MatNet** (Kwon et al., 2021) is the only neural solver runnable for general TSP to our best knowledge. **Heuristic solver: LKH** (Helsgaun, 2017) is a very strong heuristic for its high efficiency, good scalability, and impressive performance. **Learning-enhanced heuristic solver: VSR-LKH** (Zheng et al., 2021) improves LKH's heuristic strategy with RL-driven policies. The improved versions (VSR-LKH-V2/3) do not support general (asymmetric) TSP solving as we have tried. We **DO NOT** compare with the exact solver **Gurobi** since it runs out of memory for TSP instances of 100 nodes or more on our machine; and we **DO NOT** compare with **Concorde** since it does not support distance matrices as the input. Some newly proposed methods are not open-sourced yet (e.g., (Drakulic et al., 2024)) or do not provide an ATSP solver (e.g., (Drakulic et al., 2023; Ye et al., 2024b)), so we do not compare with them.

**Default Parameters.** For fairness, for LKH, VSR-LKH, and Embed-LKH, we set runs=1 (number of running iterations for each instance), max_candidates=6 (maximum number of candidates for each node). Other detailed settings (e.g., the search algorithm, the way to compute candidates, etc.) are set as the default of the original LKH program. Embed-LKH runs with 16 threads (8 for GE steps and 8 for the LKH step) as the default. On TSP-100, we set random walk window size $W = 3$ without sparsification; on TSP-100 we set $W = 1$ and adopt sparsification with $K = 100$.

**Testing Data Generation.** Before introducing data generation we should emphasize again that Embed-LKH is a one-shot solver so the training data is also the testing data. We consider the scales fo 100 nodes and 1000 nodes. The investigated distance distributions include three non-euclidean asymmetric types of distances (normal, uniform, exponential) and three types of Euclidean (metric and symmetric) distances: **i) Euclidean 2D/10D/50D.** First we generate 2D/10D/50D Euclidean random coordinates in $[0, 1]^{2/10/50}$. Then we compute the pair-wise distances, and finally re-scale them to the range of $[0, 1e4]$. **ii) Normal.** We Generate distance $\mathbf{D}_{i,j}$ from normal distribution $\mathcal{N}(0; 1)$, then re-scale them to the range of $[0, 1e4]$. **iii) Exponential.** We generate distance $\mathbf{D}_{i,j}$ from exponential distribution ($\lambda = 0.5$), then re-scale them to the range of $[0, 1e6]$. **iv) Uniform.** We generate distances from uniform distribution in $[0, 1e5]$. All distances are rounded to integers (since the float number would cause inaccurate tour lengths).

**Metrics. Average length (abbr. 'Length').** We report the average length of the found tours of all the instances. **Optimal Gap (abbr. 'Opt. Gap').** We take LKH with the largest number of trials of each scale as the reference to compute the performance gap of all compared methods. **Time.** We report the running time of the solver. For Embed-LKH, it includes the time of all the 4 steps.

## 7.2 MAIN RESULTS

We give results of TSP-100 (100nodes, 1000 instances) in Table 1 and TSP-1000 (1000nodes, 100 instances) in Table 2. We set $d = 3$ for TSP-100 and $d = 1$ for TSP-1000. Embed-LKH outperforms VSR-LKH and MatNet consistently, and We provide analysis from the following aspects:

**Observation 1: In most cases, Embed-LKH can find better solutions than LKH within the same number of trials.** It demonstrates the effectiveness of Embed-LKH's generating candidates from the ghost distances instead of the original distances.

**Observation 2: Embed-LKH is better at non-metric and asymmetric data, and also competent on high-dimensional (dimension $\geq 10$) Euclidean distances.** The gap between other methods and LKH on the normal, uniform, and exponential distances is significantly higher than on the Euclidean distances. We also observe that Embed-LKH is at the top level on high-dimensional Euclidean data. The results demonstrate the significance of GE.

**Observation 3: LKH is still competent on Euclidean data.** As shown in the results, on TSP-100, when max_trials≤1000, LKH is better than Embed LKH on Euclidean 2D data; and LKH is also superior to Embed-LKH on TSP-1000 2D Euclidean data with max_trials=10.This demonstrates the effectiveness of original LKH in 2D Euclidean problems. Theoretically, this is because the search algorithm of LKH ($\lambda$-opt) by exchanging edges in a tour has a strong physical meaning in its

Table 1: Results on TSP-100. **Bold**: methods yielding best results with the same setting.

| Setting | Method | Normal Length↓ | Opt. Gap↓ | Time↓ | Uniform Length↓ | Opt. Gap↓ | Time↓ | Exponential Length↓ | Opt. Gap↓ | Time↓ |
|---|---|---|---|---|---|---|---|---|---|---|
| | MatNet | 204521.857 | 4.753% | 2m15s | 793294.879 | 381.074% | 1m45s | 2323096.288 | 1244.204% | 6m3s |
| max_trials =10 | LKH | 196221.741 | 0.502% | 1m18s | 176221.947 | 6.865% | 1m24s | 186287.126 | 7.791% | 3m18s |
| | VSR-LKH | 197116.059 | 0.960% | 1m25s | 182244.461 | 10.518% | 1m28s | 192711.099 | 11.508% | 2m1s |
| | Embed-LKH | **195629.278** | **0.198%** | 21s | **170434.696** | **3.356%** | 20s | **179324.534** | **3.762%** | 41s |
| max_trials =100 | LKH | 195369.814 | 0.065% | 1m25s | 166124.554 | 0.742% | 1m27s | 174487.398 | 0.963% | 2m9s |
| | VSR-LKH | 199476.359 | 2.169% | 3m48s | 196072.837 | 18.904% | 2m22s | 208767.031 | 20.798% | 2m48s |
| | Embed-LKH | **195311.846** | **0.036%** | 22s | **165745.077** | **0.512%** | 23s | **173894.059** | **0.620%** | 43s |
| max_trials =1000 | LKH | 195274.407 | 0.017% | 3m17s | 165180.832 | 0.170% | 3m7s | 173104.776 | 0.163% | 6m2s |
| | VSR-LKH | 201260.070 | 3.082% | 21m19s | 206323.724 | 25.120% | 12m2s | 219028.828 | 26.736% | 11m52s |
| | Embed-LKH | **195249.138** | **0.004%** | 40s | **165018.977** | **0.072%** | 40s | **172987.579** | **0.095%** | 1m |
| max_trials =10000 | LKH | 195241.982 | – | 17m39s | 164900.749 | – | 16m18s | 172823.147 | – | 16m38s |
| | VSR-LKH | | | | Out of Time (time>3h) | | | | | |
| | Embed-LKH | **195234.123** | **-0.004%** | 2m | **164813.108** | **-0.053%** | 3m9s | **172727.728** | **-0.055%** | 3m18s |
| | | Euclidean 2D Length↓ | Opt. Gap↓ | Time↓ | Euclidean 10D Length↓ | Opt. Gap↓ | Time↓ | Euclidean 50D Length↓ | Opt. Gap↓ | Time↓ |
| | MatNet | 93240.803 | 53.309% | 1m43s | 489627.558 | 37.171% | 1m43s | 664242.653 | 3.724% | 5m57s |
| max_trials =10 | LKH | **61586.028** | **1.261%** | 1m16s | 359495.360 | 0.714% | 1m29s | 641949.717 | 0.243% | 1m31s |
| | VSR-LKH | 62861.306 | 3.358% | 1m34s | 361055.618 | 1.151% | 1m53s | 642914.388 | 0.394% | 2m14s |
| | Embed-LKH | 61598.954 | 1.282% | 40s | **358744.417** | **0.503%** | 24s | **641480.050** | **0.170%** | 22s |
| max_trials =100 | LKH | **60903.262** | **0.139%** | 1m50s | 357832.551 | 0.248% | 1m59s | 640998.944 | 0.095% | 2m14s |
| | VSR-LKH | 63106.889 | 3.762% | 3m19s | 362271.816 | 1.492% | 3m28s | 643562.573 | 0.495% | 5m11s |
| | Embed-LKH | 60910.506 | 0.151% | 34s | **357564.439** | **0.173%** | 26s | **640753.293** | **0.056%** | 25s |
| max_trials =1000 | LKH | 60823.766 | 0.008% | 7m45s | 357235.218 | 0.081% | 7m4s | 640581.210 | 0.030% | 8m23s |
| | VSR-LKH | 63163.195 | 3.854% | 56m46s | 363138.487 | 1.734% | 25m30s | 644148.481 | 0.587% | 35m21s |
| | Embed-LKH | **60822.693** | **0.006%** | 1m20s | **357076.636** | **0.036%** | 1m5s | **640450.215** | **0.009%** | 1m |
| max_trials =10000 | LKH | 60818.954 | – | 58m53s | 356947.480 | – | 46m7s | 640392.055 | – | 44m43s |
| | VSR-LKH | | | | Out of Time (time>3h) | | | | | |
| | Embed-LKH | **60818.497** | **-0.001%** | 7m41s | **356880.873** | **-0.019%** | 6m49s | **640320.954** | **-0.011%** | 6m51s |

Table 2: Results on TSP-1000, 100 instances.

| Setting | Method | Normal Length↓ | Opt. Gap↓ | Time↓ | Uniform Length↓ | Opt. Gap↓ | Time↓ | Exponential Length↓ | Opt. Gap↓ | Time↓ |
|---|---|---|---|---|---|---|---|---|---|---|
| max_trials =10 | LKH | 1826493.81 | – | 6m10s | 250579.77 | – | 7m31s | 180838.91 | – | 6m16s |
| | VSR-LKH | 1829168.70 | 0.146% | 6m27s | 245652.75 | -1.966% | 6m23s | 181457.10 | 0.342% | 6m4s |
| | Embed-LKH | **1809316.69** | **-0.940%** | 42m2s | **206660.97** | **-17.527%** | 42m13s | **174166.86** | **-3.689%** | 8m10s |
| | | Euclidean 2D Length↓ | Opt. Gap↓ | Time↓ | Euclidean 10D Length↓ | Opt. Gap↓ | Time↓ | Euclidean 50D Length↓ | Opt. Gap↓ | Time↓ |
| max_trials =10 | LKH | **176185.42** | – | 8m15s | 2450987.21 | – | 10m47s | 5570683.16 | – | 9m14s |
| | VSR-LKH | 175679.35 | -0.287% | 6m20s | 2456099.77 | 0.209% | 7m19s | 5576274.90 | 0.100% | 8m35s |
| | Embed-LKH | 176212.80 | 0.016% | 3m38s | **2445033.19** | **-0.243%** | 14m52s | **5564471.29** | **-0.112%** | 14m8s |

design for Euclidean 2D data: on a 2D Euclidean plane, the sum of the lengths of the diagonals of a quadrilateral can never be shorter than the sum of the lengths of its opposite sides. We also observe that on Euclidean 10D/50D, the gap between LKH and Embed-LKH is significantly smaller than the gap on the non-metric data, indicating LKH's advantages on the metric Euclidean data.

**Observation 4: Asynchronized Embed-LKH is efficient in most cases.** As shown by Table 1, on TSP-100, the running time of asynchronized Embed-LKH is less than most baselines. While on TSP-1000, time consumed by the additional steps of GE becomes non-ignorable. We leave the discussion of how much asynchronization improves the efficiency in different cases in Sec. 7.3.

### 7.3 PARAMETER ANALYSIS AND ABLATION STUDIES

**Window size $W$.** Experiments are conducted on TSP-1000 of the exponential distribution with a varying $W$. Results are plotted in Fig. 5 a), showing that the best window size is about $W = 3$. When $W = 1$, it means that no random walk is conducted. $W = 3$ is better than $W = 1$ indicates that a proper setting of random walk is beneficial.

**Embedding dimension $d$.** We conduct experiments on TSP-100, exponential distribution, with a varying $d$. We plot results in Fig. 5 b), showing that either a too high or too low $d$ can cause a decline in performance. $d = 3$ is the best. This also implies that simply augmenting the transition probability matrix through random walks is not enough. The dimensionality reduction step in GE is necessary.

**Number of negative samples $S$.** Experiments are conducted on TSP-100 of the exponential distribution with a varying $S$. Results plotted in Fig. 5 c) show that under this certain setting, $S = 20$

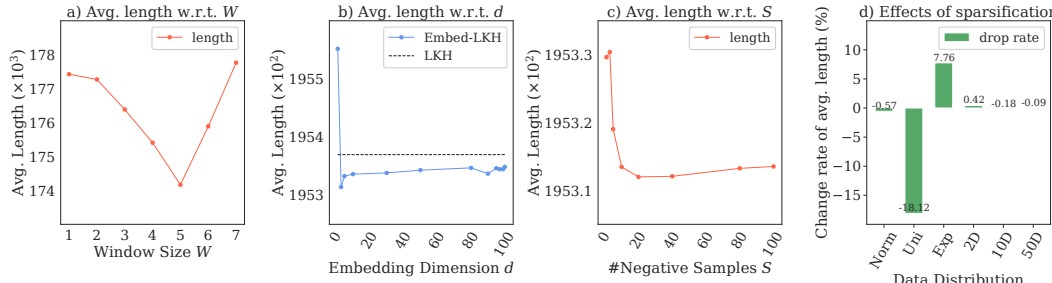

Figure 5: Ablation studies over parameters $d$, $W$, $S$, and the sparsification operation. All metrics are better when lower.

might be a good choice. However, from the absolute variation of the average length, the influence of parameter $S$ on the results is not significant when $S \geq 5$.

**Studies over the sparsification technique (in step 2).** We conduct study from the following two aspects: **i) Effectiveness on different data.** We compare Embed-LKH with and without sparsification on TSP-1000 of different distributions. We plot the percentage change in length before and after applying the sparsification technique in Fig. 5 d). The results show that sparsification

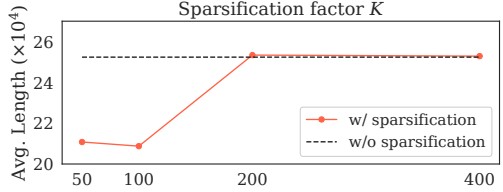

Figure 4: Parameter analysis for $K$.

has a positive effect on some data distributions (uniform, normal, Euclidean 10D and 50D) but a negative effect on others (exponential, Euclidean 2D). This implies that in some scenarios, sparsification might incorrectly mask edges that have the potential to be part of the optimal solution. **ii) Parameter analysis.** We run on TSP-1000 uniform data with different sparsification factor $K$, with results plotted in Fig. 4. Results show that when $K < 200$, the average tour length is considerably smaller than running without sparsification, demonstrating the effectiveness of sparsification in some cases and also highlighting the importance of selecting a proper $K$. **iii) Efficiency-Effectiveness trade-off.** The sparsification step may lead to significant time consumption. In our experiments of Fig. 4, we found that $K = 100$ would consumes 2533 seconds, which is more than 5 times of running without sparsification (461 seconds).

**Efficiency improvement by multi-thread asynchronization.** In Sec. 5.2 we have introduced the asynchronization of GE steps and LKH step. We run Embed-LKH with and without asynchronization and report the running time as shown in Table 3. In the table, 'Thread=2' means we use one

| Data and Setting | No Asynchorinaztion | Threads=2 | Threads=16 |
|---|---|---|---|
| TSP-100, Normal max_trials=100 | 192s | 172s | 26s |
| TSP-1000, Uniform max_trials=10 | 20574s (24x of Thread=16) | 16576s (19x of Thread=16) | 857s |

Table 3: Running time w/ and w/o multi-thread asynchronization

thread for GE steps and one for LKH; 'Thread=16' means 8 for GE and 8 for LKH. We see that when the number of threads increases, the runtime correspondingly decreases. We also observe that the acceleration brought by asynchronization is more significant on larger-scale problems (TSP-1000).

**Other findings.** We provide some interesting findings on the differences between ghost distances and original distances in Appendix E, which align well with our motivation shown in Fig. 1.

# 8 CONCLUSIONS

In this paper, we present a graph-embedding-enhanced LKH (Embed-LKH) algorithm for general TSP solving. This is the first technology to utilize graph embedding techniques to assist in solving the general TSP, and also the first learning-enhanced one that outperforms LKH on the problem of general TSP showing in the ML community, to our best knowledge.

**Limitations and Future Works.** i) The setting of parameter $d$ and the strategy $p(j|i)$ (especially the inside parameter, window size $W$) in the GE steps may require careful consideration. We leave automated parameter selection as the future work. ii) There is still a significant room for improving the efficiency of Embed-LKH by more tightly coupling the two functions `LKH_GenCand` and `LKH_SearchTour`.

## ETHICS STATEMENT

The method proposed in this paper enhances the performance of general TSP solving via the combination of neural graph embedding technique and the heuristic algorithm LKH. A dataset comprising synthetic instances of TSPs with different distributions will be released upon publication. To our best knowledge, no potential harmful impacts can be observed that need be otherwise stated.

## REPRODUCIBILITY STATEMENT

The hardware, parameters, settings, and the generation of datasets, are provided in Sec. 7.1. An anonymous repository is presented at the end of the abstract as to demonstrate basic implementation and results for reviewing. Source code and datasets shall be made public upon publication.

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

# Appendix

CONTENTS

Table 4: Main notations and descriptions.

| Notations | Descriptions |
|---|---|
| $N$ | The number of nodes in TSP |
| $\mathbf{D} \in \mathbb{R}^{N \times N}$ | Distance matrix |
| $\widehat{\mathbf{D}} \in \mathbb{R}^{N \times N}$ | Ghost distance matrix |
| $L(\tau)$ | Length of a tour $\tau$ or a path $\pi$ in TSP |
| $\mathbf{P} \in \mathbb{R}^{N \times N}$ | Transition probability transformed from $\mathbf{D}$ |
| $p(j\|i)$ | Empirical co-occurance probability that node/word $j$ shows in the neighborhood of node/word $i$ |
| $\mathbf{Q}$ | $p(j\|i)$ in the matrix form |
| $\widehat{\mathbf{P}} \in \mathbb{R}^{N \times N}, \widehat{p}(j\|i)$ | Transition probability learned by embeddings, in matrix form and scalar form respectively |
| $p(\pi)$ | Probability that a path $\pi$ (or a tour $\tau$) forms in the probability space |
| $J$ | Objective for GE |
| $\mathbf{x}_i, \mathbf{h}_i \in \mathbb{R}^d$ | Node embedding and hidden embedding of node $i$ |
| $\mathbf{X}, \mathbf{H} \in \mathbb{R}^{d \times N}$ | Node embedding and hidden embedding in the matrix form |
| $y_{ij}$ | The value of $\mathbf{x}_i^\top \mathbf{h}_j$ |
| $\mathbf{U}_d, \mathbf{V}_d \in \mathbb{R}^{N \times d}, \mathbf{\Sigma}_d \in \mathbb{R}^{d \times d}$ | Results of SVD |

## A  NOTATIONS

We list the main notations in Table 4 for reference in reading.

## B  RELATED WORKS

### B.1  TSP SOLVERS

**Exact Solvers.** Solvers for linear programming and mixed integer linear programming, e.g., Gurobi and CPLEX, can be used to solve general TSP with exact optimal solutions as output. These methods can be time-consuming in real-world applications where the scale of instances may go large. Concorde is a solver developed for TSP but it fails to deal with general TSP where only distances are available.

**Heuristic Solvers.** Some trivial heuristic algorithms include nearest neighbor (NN), furthest insertion (FI), etc. After decades of optimizations, strong heuristics such as Lin-Kernighan-Helsgaun (LKH by Helsgaun (2017)) and EAX (Nagata & Kobayashi, 2013)) have been well developed and are able to give satisfying results with good efficiency and scalability.

**Neural Solvers.** Compared with the heuristics and exact solvers, neural solvers are believed to be more efficient when the solving procedure is run parallelly on GPUs. Also, the learning-based approaches are believed to have a stronger capability in solving TSP instances from a specific known distribution. Most neural TSP solvers are developed for TSP with coordinates (Kool et al., 2018; Kwon et al., 2020; Li et al., 2023; Qiu et al., 2022; Sun & Yang, 2023; Vinyals et al., 2015; Min et al., 2024). This is due to the current lack of neural networks that can effectively learn in scenarios with **only** pairwise feature information, such as the distances in TSP. MatNet (Kwon et al., 2021) is one of the works that can be applied to BiTSP. It proposes a Transformer-based solver for asymmetric TSP (ATSP) whose input is a distance matrix. However, the heavy-encoder heavy-decoder neural architecture and the DRL-based training put limitations to its scalability and consequently its applicability to real-world problems whose scales are varying. Recently, several works (Drakulic et al., 2023; 2024; Ye et al., 2024b) conducted experiments on ATSP and showed promising results. Yet, such part regarding general TSP solving have not open-sourced.

Developing neural solvers that combine LKH and neural networks is also an important line of research. VSR-LKH (Zheng et al., 2021) improves LKH's heuristic strategy with RL-driven policies, which achieves performance improvement on 2D Euclidean TSP compared with vanilla LKH. It is a

Table 5: Technical comparison of Embed-LKH and other TSP Solvers. It should be noted that although many newly proposed neural solvers for TSP has not yet been open-sourced.

| Solver Type | Method | Distance Type | Description | Open-sourced |
|---|---|---|---|---|
| Exact solver | Gurobi | General | N/A | Available but not open-sourced |
| | CPLEX | | | |
| | Concorde | Euclidean | | |
| Heuristic solver | LKH (Helsgaun, 2017) | General | See Sec. C | ✓ |
| | EAX(Nagata & Kobayashi, 2013) | Symmetric | Genetic algorithm | ✓ |
| Neural solver | PointerNet (Vinyals et al., 2015) Ma et al. (2019) | Euclidean | GPN+RL, pre-trained | ✓ ✓ |
| | AM (Kool et al., 2018) POMO (Kwon et al., 2020) SYM-NCO (Kim et al., 2022) | | GAT+DRL, pre-trained | ✓ ✓ ✓ |
| | DIMES (Qiu et al., 2022) | | GNN+meta RL, pre-trained | ✓ |
| | GCN(Joshi et al., 2019) Att-GCN(Fu et al., 2021) | | GNN+SL, pre-trained | ✓ ✓ |
| | DIFUSCO(Sun & Yang, 2023) T2T(Li et al., 2024) | | Diffusion+Generative, pre-trained | ✓ ✓ |
| | UTSP (Min et al., 2024) QUBO (Schuetz et al., 2022) | | SAG+UL, pre-trained GNN+UL, pre-trained | ✓ ✓ |
| | BQ-NCO (Drakulic et al., 2023) GOAL (Drakulic et al., 2024) GLOP (Ye et al., 2024b) MatNet (Kwon et al., 2021) | • General | GCN+Transformer+RL, pre-trained GCN+Transformer+RL, pre-trained Divide-Conquer+RL, pre-trained Transformer+RL, pre-trained | ATSP solver not provided ✗ No ATSP solver proposed ✓ |
| Learning-enhanced heuristic solver | GNNGLS (Hudson et al., 2021) NeuralGLS (Sui et al., 2023) DeepACO (Ye et al., 2024a) NeuroLKH (Xin et al., 2021) | Euclidean | GNN+GLS, pretrained GNN+GLS, pretrained GNN+ACO, pretrained SGN+LKH, pre-trained | ✓ ✗ ✓ ✓ |
| | VSR-LKH (Zheng et al., 2021) | • General | RL+LKH, • one-shot | ✓ |
| | • **Embed-LKH** | • General | GE+LKH, • one-shot | ✓(Upon publication) |

learning-based solver yet without neural networks. NeuroLKH (Xin et al., 2021), improves LKH with the Sparse Graph Network (SGN) aimed at generating better edge candidates as a substitution for LKH's $\alpha$-nearest measure. However, SGN relies on node coordinates as input, making NeuroLKH inapplicable to general TSP.

We provide a technical comparison between Embed-LKH and all other TSP solvers in Table 5, demonstrating the significance of our work.

## B.2 WORD2VEC-BASED GRAPH EMBEDDING

We mainly discuss about Graph Embedding (GE) methods based on the language model word2vec (**?**). In this branch, GE methods have been designed for different types of networks, including homogeneous networks (Perozzi et al., 2014; Grover & Leskovec, 2016; Tang et al., 2015), heterogeneous networks (Dong et al., 2017) and multiplex networks (Liu et al., 2017; Qu et al., 2017; Xiong et al., 2021). By designing the input node sequences, GE methods achieve to preserve specific types of information, e.g., low-order proximity (LINE (Tang et al., 2015),) structure similarity (struc2vec (Ribeiro et al., 2017)), versatile similarity measures (VERSE (Tsitsulin et al., 2018)), and cross-network alignment relations (CENALP (Du et al., 2022)). Another line of GE research is developing methods based on matrix factorization. GraRep (Cao et al., 2015) solves the embedding by matrix factorization for random walk and Skip-Gram while also taking high-order proximity information into consideration. NetMF (Qiu et al., 2018) proposes to unify some word2vec-based GE methods including LINE, DeepWalk, and node2vec, etc. within a matrix factorization framework, and NetSMF (Qiu et al., 2019b) treats the network embedding task as sparse matrix factorization. AROPE and HOPE (Zhang et al., 2018; Ou et al., 2016) propose to preserve multi-order proximity by matrix factorization with some mathematical tricks. Methods based on deep neural networks (Wang et al., 2016) especially graph neural networks (Hamilton et al., 2017; Kipf & Welling, 2017; Xu et al., 2018) are also influential in literature of GE.

## C   SOME TECHNICAL KEY POINTS OF LKH

For readers not interested in how LKH works, just skip this part and regard LKH as a black box that works efficiently to find near-optimal tours for TSP. In LKH, there are three main components working in series that contribute to its strong performance,

**Step 1. Compute the $\alpha$-Nearest Measure.**   Suppose $L(T)$ is the length of the minimum 1-tree (Def. 2) of graph $G = (\mathcal{V}, \mathcal{E})$ and $L(T^+(i,j))$ is the length of the minimum 1-tree containing edge $(i, j)$, the $\alpha$-measure of edge $(i, j)$ can be calculated as $\alpha(i, j) = L(T^+(i, j)) - L(T)$. $\alpha$-measures are used for specifying the edge candidate set which consists of $p$ edges with the smallest $\alpha$-measures connected to each node ($p = 5$ as default).

**Definition 2** (**1-tree**). *Let $G = (\mathcal{V}, \mathcal{E})$ be a undirected weighted graph where $\mathcal{V} = \{1, 2, \cdots, n\}$ is the set of nodes and $\mathcal{E} = \{(i, j) | i \in \mathcal{V}, j \in \mathcal{V}\}$ is the set of edges. A 1-tree for a graph $G = (\mathcal{V}, \mathcal{E})$ is a spanning tree on the node set $\mathcal{V} \backslash \{1\}$ combined with two edges from $\mathcal{E}$ incident to node $1$. A minimum $1$-tree $T$ is a $1$-tree of minimum length.*

**Step 2. Node Penalties.**   LKH uses a subgradient optimization technique to obtain a penalty $\pi_i$ over each node $i$, then modify the distance as $\mathbf{C}_{ij} \leftarrow \mathbf{C}_{ij} + \pi_i + \pi_j$. This operation changes the distances but would not affect the optimal solution.

**Step 3. Search Solutions by $\lambda$-Opt.**   $\lambda$ edges in the current tour will be exchanged by another set of $\lambda$ edges, all of which should be included in the candidate sets of the original $\lambda$ edges, to improve the tour until no such exchanges can be found.

## D   PROOF TO PROPOSITION 2

Here we provide the proof to Proposition 2. By Eq. 4 and Eq. 9, we have

$$
\begin{aligned}
\hat{L}(\tau) &= \sum_{i=1}^{M-1} \widehat{\mathbf{D}}_{\tau_i, \tau_{i+1}} + \widehat{\mathbf{D}}_{\tau_M, \tau_1} - N \min(\widehat{\mathbf{D}}) \\
&= -\log \left( \widehat{\mathbf{P}}_{\tau_M, \tau_1} \prod_{i=1}^{M-1} \widehat{\mathbf{P}}_{\tau_i, \tau_{i+1}} \right) - N \min(\widehat{\mathbf{D}}) \\
&= -\log \hat{p}(\tau) - N \min(\widehat{\mathbf{D}})
\end{aligned}
\tag{15}
$$

by which it is obvious that $\hat{p}(\tau)$ monotonically decreases with respect to $\hat{L}(\tau)$. Then the two propositions are obviously right.

## E   OTHER INTERESTING FINDINGS

One of the questions that we are curious about is that what are the differences between the ghost distances and the original distances. Our answer is that **GE make the non-metric data more like metric data.** We investigate the TSP-100 instances of the normal distribution, we find that the number of non-metric cases ($\mathbf{D}_{i,j} + \mathbf{D}_{j,k} < \mathbf{D}_{i,k}$) decreases significantly in the ghost distance matrix after GE (from more than 10000 cases to less than 1000 cases). As the experiments show, LKH is strong at solving metric data. So, to LKH, making the non-metric data more like metric data may be beneficial to non-metric TSP solving. And that is exactly our motivation as shown in Fig. 1.

