# OpenReview forum: "From Graph Embedding to LKH: Bridging Learning and Heuristics for a Streamlined General TSP Solver"
_ICLR.cc/2025/Conference — ICLR 2025 Conference Withdrawn Submission_

### Official Review · Reviewer_rAUs · 2024-10-18

**Soundness:** 2
**Presentation:** 3
**Contribution:** 2
**Rating:** 3
**Confidence:** 4

**Summary:**

In this paper, the authors propose a hybrid algorithm for the general TSP that combines techniques used for NLP to enhance the well-known Lin-Kernighan-Helsgau (LKH) algorithm. Specifically, the authors propose a scheme consisting of four steps where (1) the general TSP distance matrix is converted to a transition probability matrix of the edges of the TSP graph, (2) the transition matrix is then processed using word2vec model to calculate X and H embeddings of the cities and edges, (3) afterward X and H are used to calculate a surrogate of the original distance matrix D named "ghost", and finally, (4) such information is used to feed the LKH algorithm.

**Strengths:**

S1. The paper combines heuristics and neural approaches to propose more efficient algorithms. I believe that as far as combinatorial optimization is concerned, this approach is the most promising for the future. In that sense, I think this work aligns well.

S2. Except for a typo (line 418), the paper is very well written and easy to follow, I enjoyed reading it.

**Weaknesses:**

W1. The main problem with this paper is that it focuses on proposing a new algorithm to solve the TSP (general in this case), and it seems that getting bold numbers in the table of results is enough. I don't see how that can contribute when (1) the instances used are not real (they are artificial) and, therefore, there is no client, entity, or company interested in the result, and (2) I have never seen a paper on a real problem for which the TSP has been used (for example, with the VRP it happens). Being aware that 50% of the optimization models developed are for the TSP, what is the purpose of doing so?

W2. The second weak point of the work is experimentation. On the one hand, they do not use solvers that we now know are super-powerful for TSP, such as Concorde. Regardless of the reasons, it is difficult to put the results obtained into context, beyond the fact that they have managed to improve the LKH. On the other hand, LKH is indeed an effective algorithm for TSP, but... within the world of heuristics there are other better alternatives with which it could be compared, such as metaheuristics, but the authors ignore this (the literature in this regard is huge).

W3. One of the strengths of the paper is that they replace the distance matrix D with the ghost distance matrix. There is something about the new matrix that makes LKH work better, but this aspect is completely overlooked in the paper. The authors leave it aside in Appendix E, but it does not satisfy their content either. What are the characteristics of the ghost matrix, and do D and LKH share the global optimum? Trying to understand this aspect would improve the contribution of this paper. I expected more from Section 6.1.

W4. Terms such as "easy" or "difficult" are not appropriate to talk about optimization instances, especially when the authors do not give a clue about their exact meaning. In optimization, we use "complexity". Using the terms of the authors, an optimization instance is "easy" or "difficult" depending on the algorithm used to optimize it. So, even if authors can define such terms, they would not serve as a general characteristic of an instance.

W5. In section 5.1, I think the term LHS was not previously defined.

W6. In line 124, the authors define tours in the TSP repeating the first city at the end of \tau. Note that with the exposed objective function, such representation of the solutions is not needed.

**Questions:**

Any progress in the weaknesses noted above would be positive for the paper. In addition, I have some questions related:

Q1. The model generates solutions in one shot. Should I understand that the model is not auto-regressive? Did the authors consider this option? Some papers in the literature suggest that auto-regressive usually is better than one-shot (obviously this is not always the case).

Q2. The authors illustrate the use of the algorithm for the TSP, however, the scientific contribution of the algorithm would be much larger if the workflow was thought for other problems and heuristic algorithms. Going beyond the TSP I think it is a value nowadays.

Q3. In the 4-step workflow proposed, I did not find a motivation to justify using the transitions probability matrix in the second step, instead of the distance matrix directly. Actually, the authors used Softmax to that end, however, there are other options. What is the effect of such transformation?

---

### Official Review · Reviewer_XxNi · 2024-10-29

**Soundness:** 1
**Presentation:** 2
**Contribution:** 2
**Rating:** 3
**Confidence:** 4

**Summary:**

In this paper, the authors propose a novel approach to generate candidate edges for LKH to solve general TSP instances based on graph embeddings. Specifically, they construct a transition probability matrix from the TSP distance matrix by applying a row-wise softmax function to the negative distances. These transition probabilities are then used to generate conditional probabilities via random walks, which the authors use to train node embeddings analogous to Word2Vec and subsequently convert these embeddings back into ghost distances. Candidate edges for LKH are then the smallest such distances. Through experiments on symmetric and asymmetric TSP instances of sizes 100 and 1000, the authors demonstrate that their approach outperforms both the standard LKH algorithm and VSR-LKH.

**Strengths:**

- The paper proposes a candidate generation method that extends beyond the traditional two-dimensional Euclidean TSP, making it applicable to general TSP instances.
- Instead of attempting to solve the TSP entirely with a neural network, the authors focus on improving LKH through learning strategies, which can also help in identifying the solver's weaknesses and understanding its behavior.
- The idea of using random walks to connect candidate edges and transition probabilities introduces an interesting concept worth further exploration.

**Weaknesses:**

The experiments are not extensive enough to determine whether the proposed approach provides a significant benefit compared to existing methods.
First, it is unclear why results are presented only for the 1000-node instances with the max_trials parameter set to 10, as the default setting is typically 1000. This choice could significantly influence LKH's performance, as demonstrated in the experiments with the instances of size 100. Additionally, the paper lacks results on real-world general TSP instances.
However, my main concern is the absence of comparisons with other candidate set generation techniques already integrated into LKH. For instance, recent works [1,2] have shown that these techniques can enhance LKH's performance in the two-dimensional Euclidean case and may already introduce the same benefit as the proposed solution.
Furthermore, I am missing an explanation of the benefits of computing the embeddings. The authors claim that a transition probability matrix based on random walks is insufficient, citing their ablation study with only one-dimensional embeddings. However, I would not expect both approaches to yield the same results, as the one-dimensional embedding $x_i^T h_j$ can only be zero if one of the two values is zero, which likely interferes with negative sampling.
It also appears that the ablation studies are conducted on different parts of the dataset, given the observed ranges of tour lengths; however, no explanation or justification for this is provided.
Overall, the paper lacks sufficient details regarding the performance of the proposed approach and under what conditions it may outperform existing candidate generation strategies. Consequently, I believe it is not ready for publication. Additionally, the paper is somewhat difficult to read due to a lack of structure and transitions between paragraphs.

[1] Heins, J., Schäpermeier, L., Kerschke, P., & Whitley, D. (2024). Dancing to the State of the Art? How Candidate Lists Influence LKH for Solving the Traveling Salesperson Problem. In International Conference on Parallel Problem Solving from Nature (pp. 100-115). Springer.
[2] Taillard, Éric D., and Keld Helsgaun. "POPMUSIC for the travelling salesman problem." European Journal of Operational Research 272.2 (2019): 420-429.

**Questions:**

How does your method perform in comparison to other candidate set generation techniques?
How does it compare to methods applicable only to metric TSP instances?
Within an instance group, how is the performance distributed? Are there specific instances where your method excels, or is it consistently superior across most instances?

---

### Official Review · Reviewer_tpiK · 2024-11-03

**Soundness:** 2
**Presentation:** 1
**Contribution:** 2
**Rating:** 5
**Confidence:** 4

**Summary:**

The paper proposes a mechanism for solving "general" traveling salesperson problems (TSP), by which the authors mean TSPs where there are no assumptions made about the distance matrix between graph nodes. The basic idea is to use a word2vec model in which the goal is to determine candidate nodes for the LKH3 algorithm through a "ghost" distance matrix generated through a graph embedding of the original distance matrix. Experimental results are given for several problem distributions in sizes 100 and 1000 with a couple baselines.

**Strengths:**

1. The authors tackle a difficult problem, one that is often ignored even though real-world routing problems often have non-Euclidean distance matrices. The ML literature has also not addressed this problem except in a couple of works.
2. At a high level, the approach proposed is fairly simple, although I note that there are many, many details both in terms of the embedding and LKH. Nonetheless, I think this can still be counted as a strength.
3. (Tentative) The experimental results seem to show improvements over the baselines. I am tentatively including this here in the interest of trusting the authors to not have made any huge mistakes, but I do have many questions about the experiments.

**Weaknesses:**

1. The paper clarity/writing quality is not great. I would rank this paper below the average submission I have seen at top conferences like NeurIPS and ICLR over the last years. The English typos are annoying, but not really the main issue. I find the propositions not well-defined and at times hard to understand. The algorithm itself is never explained. There are undefined variables (S, L() and L-hat -- these are never formally defined). See below.
2. "General TSP" is a confusing term and it ought to be removed from the paper and cleared from the paper. The generalized TSP is an established term (see, e.g., [1]) involving clusters of nodes that must be visited. The "general" TSP as a term has been rarely used. I can't find any major OR papers that use it recently, just an ML paper and a couple older works in algorithms. I believe the most "correct" term would be the non-Euclidean TSP. (Reasoning: "general" could imply many constraints, e.g., loading, time windows, etc. and that is not the focus of this work)
3. In light of the ICML paper from Xia et al. [2] I am very nervous about the results in this work. Xia et al. show that essentially the "trivial strategy" (as discussed on page 5 of this work) is often all that is really learned by the heatmap methods. This paper offers no ablation or experiments to show that this work goes beyond this trivial strategy. Even if this strategy improves the performance of LKH3, it would be completely insufficient for ICLR.

Clarity: In proposition 1, I honestly have no idea how the authors jump to having exp(-L(tau)) in the numerator. I note that do not doubt the proposition. Proposition 3 is not well-stated with a clear text that can be proven. It should be reformulated. I am not sure I really see why we should even care much about Proposition 3. The explanation in the proof is also not clear: "We find that the disturbance works in step 1" -- I have no idea what this means. Proposition 4 is interesting, but I just do not see why the paper uses this space on this proof rather than, e.g., more experiments.

Also, I find the quote from Isaac Newton on the first page completely unnecessary. Find me a paper at ICLR for which this quote is not relevant. There is nothing special about this paper that makes it necessary. Also, please tone the abstract down. I do not know what makes a paper "very novel" as opposed to just novel. I also do not know why the TSP is so "notorious". If it is notorious, it is for not even being a real problem.

[1] Pop, Petrică C., et al. "A comprehensive survey on the generalized traveling salesman problem." _European Journal of Operational Research_ 314.3 (2024): 819-835.

[2] Xia, Yifan, et al. "Position: Rethinking Post-Hoc Search-Based Neural Approaches for Solving Large-Scale Traveling Salesman Problems." _arXiv preprint arXiv:2406.03503_ (2024).

Request to the authors: Please keep your response as succinct as possible.

**Questions:**

1. What kind of Gurobi model is used to solve the TSP? It is somewhat disappointing not to have some optimal results to compare against, especially when size 100 problems are really not that large. Fischetti, Lodi and Toth were solving n=1001 ATSPs to optimality back in 2003 [1]
2. How are the runtimes in Table 1 computed? Are these CPU times? Wall times? How is parallelization taken into account in the different methods? Why does every method have a different runtime rather than a uniform timeout? I do not currently feel comfortable with the results.
3. Why do you not provide any other non-Euclidean TSP baselines? The problem has certainly been ignored recently, but the problem had plenty of interesting heuristics. Just an example, going back to 2001, a paper by Glover et al. [4]. Do we even know that LKH3 is the best algorithm for this problem? For the Euclidean TSP, LKH's quality is well-documented (although it is not the best approach there, either), whereas for the non-Euclidean TSP I am less sure. Following the references forward from [4], it just seems like there could be plenty of gems out there that are now ignored. The literature review in Table 5 is clearly not complete.

[3] Fischetti, Lodi and Toth (2003). "Solving Real-World ATSP Instances by
Branch-and-Cut" _Combinatorial Optimization—Eureka, You Shrink! Papers Dedicated to Jack Edmonds 5th International Workshop Aussois, France, March 5–9, 2001 Revised Papers_. Springer Berlin Heidelberg, 2003.

[4] Glover, Fred, et al. "Construction heuristics for the asymmetric TSP." _European Journal of Operational Research_ 129.3 (2001): 555-568.

---

### Official Review · Reviewer_7aLG · 2024-11-04

**Soundness:** 3
**Presentation:** 2
**Contribution:** 2
**Rating:** 3
**Confidence:** 3

**Summary:**

The paper presents a modification to the Lin-Kernighan-Helsgaun (LKH) heuristic for the general traveling salesperson problem (TSP). The authors propose a new approach to transform an instance (i.e., the distance matrix) into a simplified version that maintains the same optimal solution as the original. They achieve this by generating a modified distance matrix using graph embedding techniques. Notably, this approach requires no training phase, allowing it to be applied directly to test instances without additional model training. The authors enhance performance by parallelizing their method, leveraging 8 threads for computation. Their results indicate that the modified LKH algorithm performs better than the original on asymmetric instances

**Strengths:**

- A key strength of this paper is its focus on asymmetric TSP instances, which do not adhere to the triangle inequality. This contrasts with most ML methods that often assume symmetric instances. This contribution is significant because many real-world routing problems exhibit asymmetries, making this approach more applicable to real-world scenarios.

**Weaknesses:**

- **Limited Scope of Application**:
The proposed method is limited to the TSP, which is among the simplest of such problems and quite distant from the complex problems encountered in real-world scenarios. For instance, real-world routing problems often involve constraints like multiple depots, time windows, or vehicle capacities, which are not addressed by the TSP. It remains unclear whether this approach could be generalized to tackle these more complex, real-world problems. To demonstrate the broader impact and practical relevance of their work, the authors should extend their evaluation to other routing problems.
- **Method Description**:
I found the paper very difficult to understand. While the introduction provides a high-level overview, the methods section jumps into dense mathematical detail with limited explanatory steps in between. This lack of intermediate, intuition-driven explanation makes it difficult for readers.
- **Unfair Comparison**:
The authors compare the modified LKH method, which is heavily parallelized using 8 threads, to the original LKH, which runs on a single thread. This difference in computational resources creates an unfair comparison and may exaggerate the advantages of the proposed approach. A more balanced evaluation, with both methods operating under equivalent computational resources, would provide a clearer picture of the true performance gains of the modified method.
- **Limited Performance Improvements**:
While the proposed method does outperform the original LKH in most cases, the performance gains are often marginal. To my understanding LKH has been mostly designed with the 2d euclidean instances in mind, it is unclear whether similar improvements could be achieved using simpler adjustments, such as parameter tuning or minor algorithmic enhancements. Providing a comparison against a fine-tuned version of LKH on asymmetric instances would help to contextualize the effectiveness of the proposed modifications and clarify whether they represent a substantial improvement over existing methods.
- **Placement of Related Work**:
The authors have moved the related work section to the appendix. Given the overall scope of the proposed method and the extended page limit of 10 pages this should not be necessary.

**Questions:**

.

---

### Note · Authors · 2024-11-29

**Comment:**

Thanks for all reviewers' time and their valuable suggestions. We would continue to improve the paper according to the reviews.

**Withdrawal Confirmation:**

I have read and agree with the venue's withdrawal policy on behalf of myself and my co-authors.